# Comparison of Physical and Thermal Properties of Mulching Films Made of Different Polymeric Materials

**DOI:** 10.3390/ma15217610

**Published:** 2022-10-29

**Authors:** Tomasz Stachowiak, Przemysław Postawa, Krystyna Malińska, Danuta Dróżdż, Agnieszka Pudełko

**Affiliations:** 1Department of Technology and Automation, Faculty of Mechanical Engineering and Computer Science, Częstochowa University of Technology, Al. Armii Krajowej 21, 42-201 Częstochowa, Poland; 2Faculty of Infrastructure and Environment, Czestochowa University of Technology, Brzeznicka 60A, 42-200 Czestochowa, Poland

**Keywords:** biodegradable polymers, mechanical properties, thermal analysis, mulching films

## Abstract

The development of polymer materials causes their huge expansion into various areas of everyday life, as well as plant and animal production. Their chemical resistance, good physical properties, and ease of processing result in an increasing use of this group of materials. Outdoor plant production both in open plantations and greenhouses requires various types of materials supporting the vegetation process as well as protecting against pests and weeds. A large group here are various types of materials used for covers of field crops, the main role of which is to prevent uncontrolled and excessive growth of weeds and thus reduce the use of herbicides as plant protection products. Cover films also have other important functions, such as reducing direct water evaporation, better moisture retention around the root system, increasing soil temperature (faster vegetation), etc. However, as always, the problem of introducing new material into agriculture production and the difficulty of its disposal arises. In recent times, farmers’ interest in various forms of mulch to protect crops and increase yields has grown significantly. In the publication, the authors attempted to analyze selected commercial properties, but also mulch produced on a laboratory scale, based on biodegradable and petroleum-derived materials.

## 1. Introduction

Polymer materials are the most popular human processed materials. In terms of volume, their global production exceeded metal production 30 years ago. The reason is the unique physical properties that give them very wide application areas. They can be used in every area of life and economy, from simple packaging applications to machine parts. Polymer materials are also successfully used in modern and sustainable agriculture, both in plant and animal breeding. In the group of agricultural products, films produced with the use of free-blow extrusion technology, used mainly as crop cover materials and tunnels, have a large share [1,2,3,4,5,6].

The vast majority of agricultural films were made mainly of classic polymers, which are produced from crude oil. The use of these films in agriculture is a serious problem from an ecological point of view. After the vegetation process, these materials contain huge amounts of plant and soil pollutants, which makes recycling difficult and causes a significant decrease in the profitability of recycling. High energy and fuel costs result in the escalation of costs related to recycling (energy, transport, waste disposal, water, detergents), which in consequence is the reason why farmers often use the practice of burning this type of waste directly on farmlands.

In recent years, there has been an increasing emphasis on the development of organic farming, the main goal of which is to minimize or completely eliminate the use of pesticides and herbicides, and also to reduce soil contamination with waste (including micro-plastics from the use of non-renewable materials). Consequently, the use of polymer materials obtained and produced from renewable sources [6,7], which under the influence of appropriate environmental factors (UV light, oxygen, moisture, bacteria) can decompose into simple, low-molecular substances that do not pose a threat to the natural environment [8,9]. Cover films are among the polymeric materials characterized by the highest degree of soiling, which entails significant costs of washing them for re-development or disposal. The use of bio-derivative and biodegradable polymers is the simplest and shortest way to eliminate the problem of land-filling and re-management of agricultural waste from mulching film [10,11,12,13,14,15]. Many researchers take up the subject of biodegradable films and modern polymer compositions and polymer materials that can become substitutes for the classic polymer materials (petroleum polymers). Ning et al. [10] discussed the topic of using biomass materials for the production of biodegradable films. The conducted research showed the possibility of producing this type of film and allowed to determine the required composition and composition of the film (the obtained films belonged to the group of single-layer films). Most of these materials are experimental products, not currently processed on an industrial scale. Pirsa et al. [11] presented the use of bioproteins in the preparation of biodegradable films and polymers. The results turned out to be promising and particular attention should be paid to good oxygen transfer and good mechanical properties. Gouveia et al. [12] described the process of obtaining biodegradable films from pectins. The obtained materials were formed into a finished product using the thermo-compression process (the process of making this type of film is also a process on a laboratory scale). The presented process was defined as an alternative to the production of eco-friendly, bio-based plastic materials.

Currently, nearly 370 Mg of plastics are produced each year—all of these materials are made of crude oil and, as already mentioned, are not self-decomposing materials. About 55 million Mg of thermoplastics are produced and processed in the EU each year [1]. The consumption of classic polymer materials in agriculture is constantly increasing. In 2017, nearly 3 Mg of thermoplastic polymeric materials were used to produce mulching films, and their consumption is forecasted to be increasing. Unfortunately, along with the increasing amount of polymer materials used, there is a problem with waste management [6,7].

Biodegradable plastics were introduced to the market in the 1980s and the biodegradable materials used at that time belonged to the so-called first-generation and could be used in less demanding applications, e.g., in the packaging industry. Modern biodegradable polymers (3rd generation) are materials that are used not only in the packaging industry but also in other industries [13,14,15,16,17,18,19,20,21].

In modern agriculture, single or two-layer mulch films made of bio-based and biodegradable polymers are more and more often used as an alternative to those produced from non-renewable sources (crude oil) [10,11,12,13,14,15,16,17,18,19,20,21]. There is a significant progress on the use and implementation of experimental materials that have a huge degradation potential (with appropriate composting conditions) and are produced from bio-based raw materials [22,23,24,25,26,27,28,29,30].

The article develops the subject of production bio films from commercially available biodegradable materials (based on commercial polymeric bio-based and biodegradable materials, which are made of potato starch). A literature review showed that currently produced biodegradable films are mainly single-layer and double-layer films produced in laboratory conditions. As part of the publication, an attempt was made to design multilayer films (at least three-layer), in which one or more layers would be functional layers, containing fillers, fertilizers, and plant protection products. Biodegradable film is a crop cover material that should disintegrate within one season. Biodegradable covering films can be active materials that have a positive effect on the properties of the soil and modify its properties, e.g., by protecting the crop against parasites or releasing nutrients, insecticides, etc. as a result of decomposition in the soil.

The aim of the publication is a comparative analysis of the properties of conventional films (made of crude oil) and films made of renewable raw materials and at the same time biodegradable in terms of mechanical and thermal properties related to the application. Selected properties were analyzed for three groups of cover films: films and fabrics made of polyethylene and polypropylene, commercially available biodegradable films, and experimental, multi-layer biodegradable films.

## 2. Materials and Methods

The foils most frequently used in agriculture as covers for plants and flowers were selected for the research. In the conducted research, a number of film properties were analyzed, including mechanical and thermal properties, and the results of one of the three-layer films in terms of biodegradation under industrial and home composting conditions were presented.

### 2.1. Tested Materials

Table 1 presents the basic features and properties of the tested materials with their trade names. Samples marked as CUT1, CUT2, and CUT3 are three-layer samples made non-commercially as part of the implementation of the Horizon 2020 project with the acronym Organic + (Table 1). Mulching films from two sources were analyzed. The first of these were commercially produced classic films (Table 1):

These films were produced on the basis of fossil, petroleum-derived raw materials. The second group consisted of biodegradable films. Biodegradable films have also been divided into two groups, i.e., commercial biodegradable films (which include films (Table 1)). All the CUT series films prepared had the ABA structure (multilayer structure) and the properties of the base polymers forming. Structure of the layers presented in Table 1. The biodegradable polymers Bioplast 400 ELIT and Bioplast 400 D were used to produce three types of multilayer films (Table 2).

### 2.2. Technologies Used to Manufacture the Tested Films

Free extrusion blow moulding technology was used for preparing CUT samples. Samples were prepared using the laboratory extrusion line in MARMA Polskie Folie Sp. z o.o. (Rzeszów, Poland) equipped with two extruders (first extruder for preparation of inner layer and the second for outer layers). Each extruder was equipped with a gravimetric dosing system for accurate dosing of used fillers (calcium carbonate and pigment). The stages of production were presented in Figure 1. Because the technology is not the main aim of the publication and samples were produced in outer company the processing conditions were not presented.

### 2.3. Analysis of Selected Mechanical Properties of Films

The study of the mechanical properties of the films used during the project included the determination of parameters such as:−maximum stress determined in the static tensile strength test,−maximum deformation (percent deformation) of the film during the static tensile strength test.

The determination of the mechanical and strength parameters of the film was carried out in accordance with the following standards: ISO 527-1 Plastics—Determination of tensile properties—Part 1: General principles, ISO 527-3 Plastics—Determination of tensile properties—Part 3: Test conditions for films and sheets [27,28,31,32]. The methods are used to investigate the tensile behavior of the test specimens and for determining the tensile strength, tensile modulus, and other aspects of the tensile stress/strain relationship under the conditions defined. During the test, the specimen is extended along its major longitudinal axis at a constant speed until the specimen fractures or until the stress (load) or the strain (elongation) reaches some predetermined value. During this procedure, the load sustained by the specimen and the elongation were measured. The universal testing machine type EZ-LX equipped with 1500 N force cell by Shimadzu Comp. (Tokyo, Japan) were used in the tests. The testing machine presented in Figure 2 with the handles was used to determine the mechanical properties of the tested conventional and biodegradable films.

The determination of the mechanical properties during the static tensile test was carried out in order to verify the application properties of the film, which include film strength and elongation at break. All films for mulching crops are applied (unfolded) on the soil with the use of agricultural machinery, which is why is their breaking strength and breaking elongation are such important features, the high value of which guarantees trouble-free application in the field. The relationship of mechanical properties of the stress-strain type is one of the basic performance parameters that define the properties of materials and the possibility of their use in selected applications. The used testing machine is equipped with self-locking handles covered with hard rubber, which guarantees correct measurement and its repeatability, which is very important when testing materials with a slippery surface and at the same time small thickness. Such a surface finish of the jaws guarantees a real measurement of the breaking force and prevents the tested films from being cut at the point of contact with the sample holders.

### 2.4. Determination of Impact Resistance by the Free-Falling Dart Method

The determination of the puncture resistance of the films was carried out in accordance with the standard ISO 7765-1:2005, Plastics film and sheeting—Determination of impact resistance by the free-falling dart method—Part 1: Staircase methods [27,28,33]. Original research stand (made on the basis of the cited standard) designed and manufactured at the Czestochowa University of Technology was used for the study (Figure 2). A measuring dart with a diameter of 38 mm and a set of additional weights with a total weight of 195 g were used. Measuring dart was dropped from a height of 1500 mm. Before starting the measurement, the samples were placed in a special measuring holder (Figure 2), consisting of two rings with an internal diameter of 125 mm and an external diameter of 150 mm (they were secured with rubber gaskets with a hardness of 60° Shore preventing the samples from slipping and moving during the measurement) and clamps. Before starting the measurement, dart with a mass of 108 g was placed in the dart holder and the electromagnet was activated in order to immobilize the dart. After preparing the stand, the dart was released, and result was recorded. The measurement results were recorded as the value of the film breakthrough or the lack of its damage, moreover, the mass of the impact damage was determined.

### 2.5. Differential Scanning Calorimetry (DSC)

The tests were carried out with the use of the DSC Polyma 214 device made by NETZSCH^®^ (Selb, Germany) with measuring possibilities in the temperature range −200 °C ÷ 600 °C with a heating rate from 0.1 to 50 K/min. The negative values of the temperature were obtained by introducing liquid nitrogen to the measuring chamber. The tests were carried out in accordance with the standard: ISO 11357: Plastics-Differential scanning calorimetry (DSC) Part 1: General principles [34]. According to the standard, aluminum crucibles (Al) were used in the tests and an atmosphere of inert gas (nitro-gen) was used. The instrument was turned on an hour before the start of the actual measurements. DSC measurements on polymers are greatly affected by the thermal history and morphology of the specimen, it is recommended that the heating or cooling run be carried out twice. In this case we ran our test only once because the aim of the research was to determine the thermal properties of the produced films and not the material from which they were produced. The first run reflects the as-received state and is performed up to the melting or glass transition where the material reaches thermal equilibrium. The samples were tested according to the following temperature program: cooling to 0 °C, isothermal stabilization 5 min, heating up to 200 °C, isothermal stabilization 5 min, cooling to 0 °C, isothermal stabilization 5 min, second heating to 200 °C, end of measurement.

### 2.6. Analysis of the Gloss of the Tested Films

The gloss measurement was carried out in accordance with ISO 2813: 2014, Paints and varnishes—Determination of gloss value at 20°, 60°, and 85° [35]. Gloss measurement was carried out using a 3nh GLOSS METER with three measuring geometries glossmeter, made in China (Nanshan District, Shenzhen, China). The measurement was carried out at room temperature and 50% humidity.

## 3. Results

### 3.1. The Results of the Analysis of the Mechanical Properties of the Tested Films

Tensile strength tests were carried out in 3 replications for each of the tested samples, and then the average was drawn and presented in collective bar graphs. The results were divided into two groups relating to the obtained values: the maximum deformation and the maximum stress of the tested samples [27,28,31,32]. The comparison of the obtained results is presented in Figure 3 and Figure 4.

The highest stress value is shown by the commercial Mypex sample (it should be noted, however, that it is a fabric made of polypropylene fibers) with a much greater thickness than the other tested samples, but due to the material used, it had a relatively low strain at break (60.5%). With this type of material, the loads are several times greater than that of other tested materials. On the other hand, when comparing all the other films, both from renewable sources (biodegradable BD) and non-renewable PE, the highest stress value was obtained for the multi-layer non-commercial CUT1 film (18.7 MPa). Slightly lower values, oscillating around 13 MPa, were recorded for the remaining non-commercial samples CUT2 and CUT3 and the commercial Polythene. The remaining commercial biodegradable films showed much lower stresses at break (approx. 8 MPa).

The results of elongation at break are very interesting. For non-commercial three-layer biodegradable samples (CUT1 and CUT3), nearly three times higher elongation values were recorded, reaching over 355% for the CUT3 sample and 300% for the CUT1 sample. The CUT2 sample obtained only 73 MPa, which was most likely caused by the addition of calcium carbonate to the middle layer, which during stretching caused delamination and faster propagation of the crack, transferring it to the surface layers. Surprisingly low elongation values were obtained for the remaining samples, except for the Mypex sample, which is a disadvantageous feature from the point of view of their mechanical application in field conditions with the use of agricultural machinery. A gentle tug can cause loss of continuity.

### 3.2. The Results of Determination of Impact Resistance by the Free-Falling Dart Method

The determination of impact resistance by the free-falling dart method was carried out on the original research stand, made in accordance with the standard ISO 7765 [33]. Table 3 presents the obtain results.

For each measurement, 10 repetitions were made (*n* = 10). The semicircular dart in diameter and weighing 108 g was drop from a height of 1500 mm onto the sample placed in a special sample holder with an internal diameter of 125 mm. The purpose of the measurement, in accordance with the assumptions of the standard, was to determine whether a sample of the tested film is punctured or not. For the tested samples, it was shown that the CUT1, CUT2, Biofilm, Polythene, and Biopolyane films were destroyed (the film was punctured by the falling dart) using the weight of the dart itself. The weight of the impact load determined on the basis of the standard was 95 g for these films. Damage due to an impact of the dart. The measuring mass has been increased by 26 g (additional load). Due to the weight of the dart and the additional weight, the CUT3 foil was destroyed (the measurement was repeated nine times). It was proved that for the CUT3 foil the impact load weight was 115.8 g. The Mypex foil was tested from the minimum load (only dart) to the use of all additional weights, despite the use of an increased mass, the foil was not damaged (the mass of the impact load could not be determined). As in the case of the static tensile strength test carried out in accordance with ISO 527, similarly, in this case, Mypex and CUT3 films have been shown to have better strength parameters, this applies to both static and dynamic tests (it should be noted, that Mypex film is a much thicker fabric than other films).

### 3.3. Thermal Properties Analysis Results

The results of the thermal analysis using the DSC method are presented in the form of thermograms in the temperature range from 0 to 200 deg. C. For better readability and comparison, pooled thermograms of commercial samples (Biopolyane, Biofilm, Polythene, and Mypex) were separated from non-commercial samples (CUT1, CUT2, CUT3). Thermograms were edited and calculated using Proteus 8.01 software. With its use, basic values were determined for each distinct endothermic peak:−peak temperature value,−enthalpy of melting transformation,−peak width and height,−onset and the end of the transformation.

The thermal properties determine, on the one hand, the processing properties of the materials used for the films, and, on the other hand, show the nature of the changes taking place in them, suggesting the composition of individual materials, and for the CUT series samples also the influence of additives on the thermal effects occurring in these samples.

Figure 5 shows that Biofilm and Biolane samples have flat and very wide peaks of melting changes, which is characteristic of biodegradable plastics made of organic raw materials. The thermograms for films made of non-renewable raw materials (Polythene and Mypex) look different. According to the manufacturers’ information, peaks characteristics of polyolefin materials (PE and PP) are visible. The Mypex sample has a distinct peak associated with the melting of the crystalline phase characteristic of polypropylene, as evidenced by the temperature of 164 deg. C. On the other hand, the Polythene sample has a wide melting range with three distinct sub-peaks at the temperature of 46.7 deg. C, 111.5 deg. C, and the largest 124.4 deg. C. This may indicate the presence of two types of polyethylene (low molecular weight polyethylene and linear polyethylene, as well as low molecular weight additives resulting in greater flexibility).

For the biodegradable film Biopolyane, the occurrence of three exothermic transformations was recorded at the temperatures of 46.8 °C, 113.2 °C, and 148.6 °C, respectively. The transformations take place at significantly different temperatures, each of which corresponds to the presence of a different type of material or polymer. Apart from the characteristic temperatures, the transformations differ significantly in the obtained enthalpy. Only the peak defined as the main one for which the transformation (melting) takes place at the temperature of 113.2 °C is characterized by the enthalpy value above 10 (J/g). The other two transformations were recorded and assigned and the values, however, they are not greater than one. The Biofilm film is characterized by the occurrence of two exothermic transformations which were defined at 99.0 °C and 158.8 °C, respectively. The transformation obtained at the temperature of 99.0 °C is significantly characterized by the value of the enthalpy of fusion (68.38 J/g).

Figure 6 shows the collective thermograms of non-commercial biodegradable samples (CUT1, CUT2, and CUT3). The occurrence of many peaks/exothermic changes was observed for them. CUT1, CUT2, and CUT3 films are characterized by similar DSC thermograms. For each of these materials, several exothermic changes were recorded during the heating of the samples. The CUT1 film is characterized by three exothermic transformations occurring successively at the temperatures of 49.0 °C, 114.4 °C, and 159.8 °C. These transformations are characterized by a low enthalpy value (the enthalpy values for the transformations are similar). The DSC course for the CUT2 sample also showed the occurrence of three exothermic changes occurring successively at the temperatures of 57.9 °C, 124.2 °C, and 158.8 °C. Note that the sample CUT2 enthalpy values are significantly higher than the sample CUT1 enthalpy values. For the CUT3 sample, four transformations and the corresponding melting points were recorded, successively they are 54.3 °C, 126.0 °C, 153.3 °C, and 173.3 °C. For the CUT foil, the differences in the obtained thermograms may result from the fillers used and (for the CUT3 foil) the different polymer composition and the addition of a dye. The probable reason for “increasing” the value of the melting enthalpy of all transformation ranges were the additives used, which contributed to a different (faster) crystallization kinetics taking place in the polymer, becoming a heterogeneous nucleus of the crystallization process. The peak for the CUT3 sample also shifted towards higher values, reaching 173.3 °C. The onset of transformation in the samples from the CUT series is similar to that for commercial biodegradable samples and falls on about 25 °C.

It should be also pay attention to the differences in the values of the enthalpy of melting of classic films (Mypex, Polythane) and the analyzed biodegradable films (CUT FILMS, Biopolyane, Biofilm) (Figure 7). The enthalpy of transformations of the characteristic biodegradable films is significantly lower than that of conventional (petroleum derivative) films. This means that for biodegradable films, much less energy is required/needed for the destruction of the ordered phase than for classic films, which translates into energy consumption, easier processability, and shorter plasticizing time.

### 3.4. Results of Determination of Gloss Measurement

Table 4 present the results of the measurement of the gloss of tested films [35]. Gloss measurement was carried out with three measuring geometries glossmeters. The table shows the average values of five measurements.

The obtained results allow us to conclude the highest gloss value for CUT1 and CUT3 films, and the lowest gloss value for Biofilm and Biopolyane films. The CUT1 film is a white film with a high light transmittance coefficient (conclusion based on visual analysis), while CUT3, Biofilm, and Biopolyane films are black, non-transparent films (visual analysis). The significantly lower gloss values for the CUT2 film are due to the content of calcium carbonate as the filler. A higher value of the gloss of the covering films may translate into an increased light reflectance and, consequently, a positive effect on the storage of moisture in the soil. Biofilm and Biopolyane films are matte films with a low gloss value, which may indicate excessive accumulation of energy (temperature rise) and faster evaporation of moisture accumulated in the soil (requires further research).

## 4. Conclusions

The conducted tests and analyzes have shown that the mechanical properties of selected biodegradable films (CUT and Biofilm series films) do not differ significantly from the properties of commercially used mulching films made of materials that are not subject to the biodegradation process.

Foil with a woven structure, including Mypex, made of PP (polypropylene), has a much greater mechanical strength, which guarantees their long-term use, which is why they are most often used for perennial covers and often contain additives protecting them against the destructive effects of UV radiation.

The great advantage of biodegradable and bio-based films is, above all, the possibility of obtaining them from renewable sources, with much lower energy expenditure, as well as the reduction of the carbon footprint and the possibility of degradation in soil or specially prepared composting installations. During their use, you can opt out of their recycling, which involves the need to collect them from the field, temporary storage, compact, transport to a recycling company, wash, and granulate, as is the case with films from fossil raw materials (PE or PP). Biodegradable films from renewable sources can be left in the field, mixed with the soil during field work and allowed to biodegrade. It should be remembered that such a way of handling the residues from the vegetation process will be longer than subjecting biodegradable films to the industrial composting process.

The conducted research has shown that it is possible to produce multi-layer biodegradable films containing various types of functional fillers. The three-layer films (CUT1, CUT2, CUT3) obtained in the extrusion blow molding process are in many cases characterized by better mechanical properties than their counterparts made of petroleum raw materials. Due to their origin, CUT foils are also characterized by different thermal properties (significantly lower melting enthalpy), which translates directly into their processing and energy consumption (lower melting enthalpy means less energy required to melt and plasticize the material). CUT3 foil (excluding Mypex foil) showed the best mechanical properties (static and dynamic). In addition, the CUT3 film is characterized by high gloss, the relatively high reflectivity of the film may have a positive effect on the amount of moisture retained in the soil (this may have a positive effect on the development of crops). CUT films are obtained from commercially available biodegradable raw materials (produced on an industrial scale), and industrial installations were used for their production.

The research showed very promising physical properties of non-commercial CUT samples from the point of view of their application, which puts them in a better position compared to commercial films for mulch plants. Of course, their degradation properties should also be compared during field tests. The results of these studies will be presented in subsequent publications of the team.

## Figures and Tables

**Figure 1 materials-15-07610-f001:**
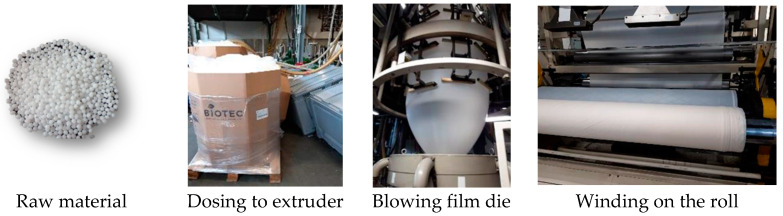
Raw material used for CUT samples preparation and view of extrusion stages.

**Figure 2 materials-15-07610-f002:**
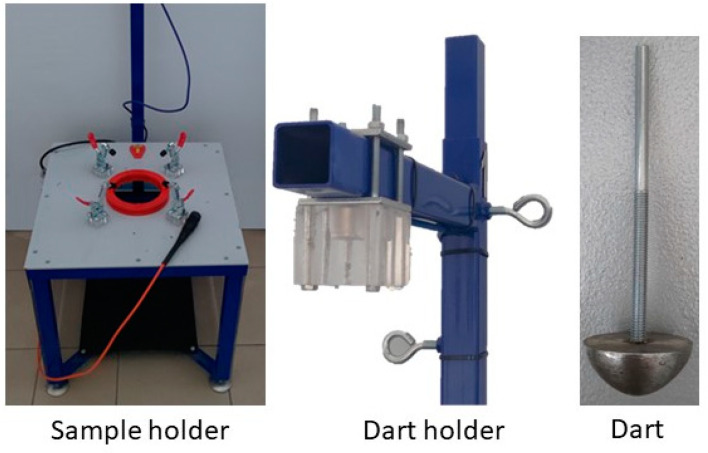
Original research stands of impact resistance by the free-falling dart determination.

**Figure 3 materials-15-07610-f003:**
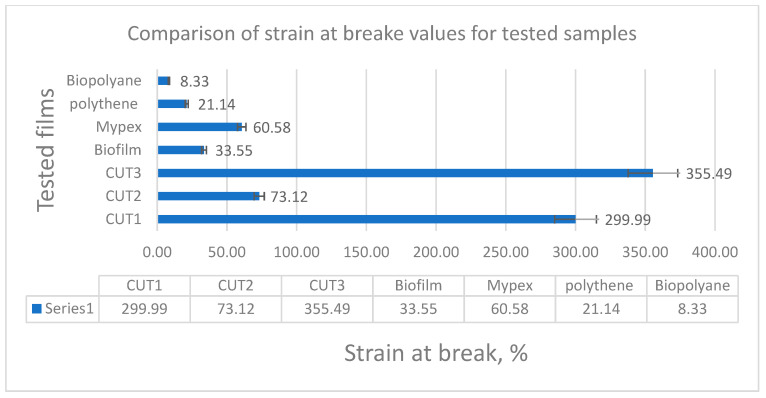
Comparison of the maximum strain at brake of the tested samples.

**Figure 4 materials-15-07610-f004:**
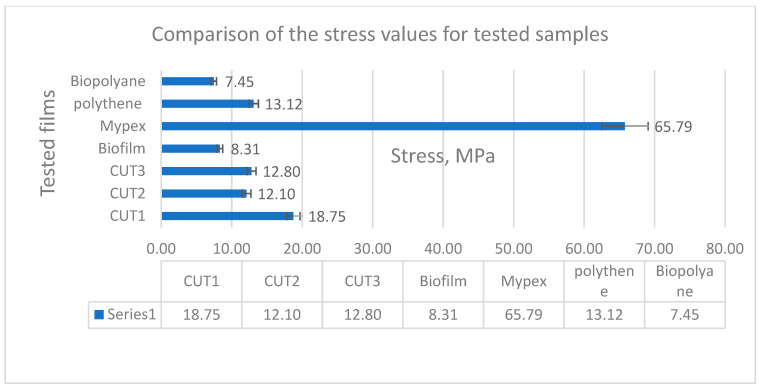
Comparison of the maximum stress obtained in the static tensile strength test.

**Figure 5 materials-15-07610-f005:**
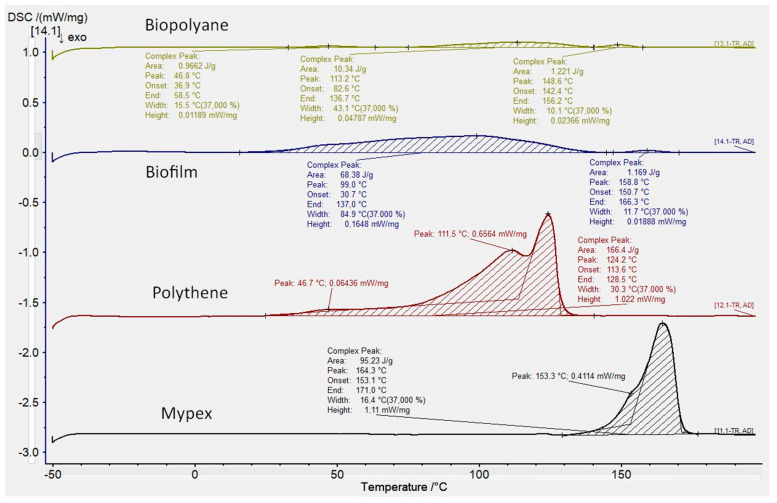
DSC thermograms obtained for samples of conventional and commercial biodegradable films.

**Figure 6 materials-15-07610-f006:**
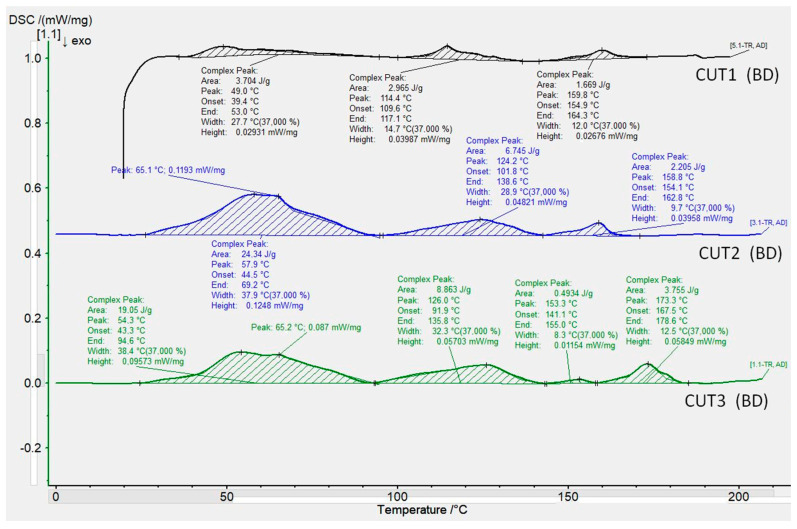
DSC thermograms obtained for CUT foil samples.

**Figure 7 materials-15-07610-f007:**
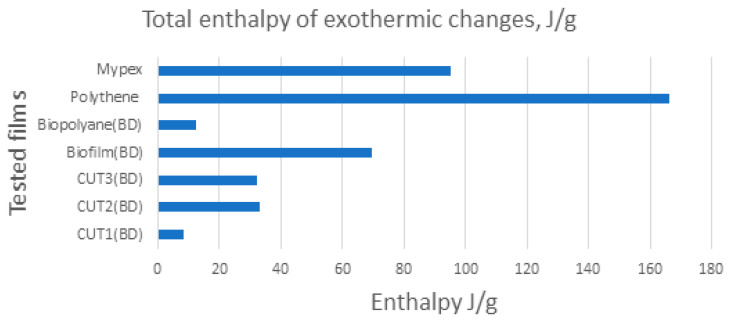
Total enthalpy of exothermic changes taking place in the studied samples.

**Table 1 materials-15-07610-t001:** Commercial, biodegradable and non-biodegradable mulching films used in the research.

Commercial Non-Biodegradable Cover Films		
Sample Name	Properties Description	View of Sample
Mypex	Woven mulch fossil-based polypropylene, Thickness: 0.25 mm, Colour: Black. Commercial product. Non-biodegradable	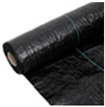
Polythane	Thickness: 0.04 mm. Colour: Black, glossy. Commercial fossil-based polyethylene. Non-biodegradable	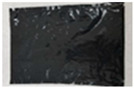
Biofilm (BD)	Thickness: 0.015 mm. Colour: Black, mat. Commercial non fossil based. Biodegradable	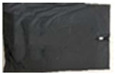
Biopolyane(BD)	Thickness: 0.015 mm. Colour: Black, mat Commercial non fossil based. Biodegradable	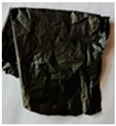
CUT1 (BD)	Thickness: 0.025–0.030 mm Colour: transparent, glossy, structure ABA Layer A—Bioplast 400 ELIT longer time of degradation Layer B—Bioplast 400 D faster degradation without filler	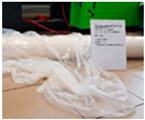
CUT2 (BD)	Thickness: 0.030–0.035 mm Colour: semi-transparent, glossy, structure ABA Layer A—Bioplast 400 ELIT longer time of degradation Layer B—Bioplast 400 D faster degradation + 20% of calcium carbonate filler CaCO3 (biobased from sea shells)	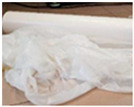
CUT3 (BD)	Thickness: 0.038–0.042 mm Colour: black, non-transparent, glossy, structure ABA Layer A—Bioplast 400 ELIT longer time of degradation Layer B—Bioplast 400 D faster degradation + 5% BLACK colourbatch FDM 85911 BK BIO1 MASTERBATCH-PolyONE certificated bio based and biodegradable black pigment	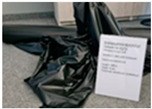

**Table 2 materials-15-07610-t002:** Basic properties of biodegradable polymers used to produce CUT foil.

Parameter	Bioplast 400 ELIT (Inner Layer B)	Bioplast 400 D (Outer Layer A)
Density	1.16 g/cm^3^	1.28 g/cm^3^
Bulk density	745 kg/m^3^	820 kg/m^3^
Mass Flow Rate (MFR) (190 °C/5 kg)	12.4 g/10 min.	7.5 g/10 min.
Moisture content	less than 0.3% weight	

**Table 3 materials-15-07610-t003:** Results of determination of impact resistance by the free-falling dart method.

Film Name	Dart	Dart + Initial Weight	Dart + Max. Weight	Impact Damage Mass—mf [g]
**CUT1**	foil puncture	-	-	**95**
**CUT2**	foil puncture	-	-	**95**
**CUT3**	no puncture of the foil	foil puncture		**115.8**
**Biofilm**	foil puncture			**95**
**Mypex**	no puncture of the foil	no puncture of the foil	no puncture of the foil	**no puncture of the foil**
**Polythene**	foil puncture			**95**
**Biopolyane**	foil puncture			**95**

**Table 4 materials-15-07610-t004:** Gloss measurement results for geometries 20°, 60°, and 85°.

	20°	SD	60°	SD	85°	SD
**CUT1**	4.86	0.25	34.56	0.84	45.76	1.50
**CUT2**	1.96	0.06	13.43	0.47	20.53	1.48
**CUT3**	2.4	0.26	33.93	1.12	50.03	2.47
**Biofilm**	0	0.00	0.3	0.10	4.5	0.26
**Mypex**	0.16	0.06	5.9	0.92	6.06	0.64
**Polythene**	1.1	0.30	15.73	1.75	34.26	3.55
**Biopolyane**	0	0.00	0.33	0.21	2.36	1.65

## Data Availability

Not applicable.

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
