# Peer review of "Comparison of Physical and Thermal Properties of Mulching Films Made of Different Polymeric Materials"

_materials, 2022, doi:10.3390/ma15217610_

Round 1

Reviewer 1 Report

The presentation of the charts needs to be improved

The references should be more recent

The analysis of the data needs to be enhanced with more evidence 

The novelty is not highlighted in the paper

The methodology needs more clarification

Author Response

Reviewer 1 - Answers

Publication title :

“Comparison of physical and thermal properties of mulching  films made of different polymeric materials”

System number : materials-1982982

Dear Reviewer

At the outset, I would like to thank you for the prepared review. I agree with all the comments. The answers have been included in a separate file which I am attaching. 

Reviewer 2 Report

Dear Authors,

I studied your manuscript entitled "Comparison of physical and thermal properties of mulching films made of different polymeric materials". This research isn't conducted correctly. Additional experiments are needed to improve the manuscript in terms of journal quality. Thus, I do not recommend the publication of your hastily written paper in the Materials journal without a deep revision.

1) The quality of the abstract and conclusion should be enhanced by including significant research findings. More quantitative data in these sections would be beneficial.

2) The recent literature review should be summarized for benchmarking purposes and discussed in detail with your research findings.

3) You should present some more analyses that evaluate the properties of the used films. Therefore, the results of additional analyses such as barrier properties, tear resistance, tensile strength, optical properties, degradation properties, microscopy analysis, etc. should be presented and discussed.

4) The references started with (1, 20-24) in the manuscript, please renumber your references. No reference is also given in paragraph 2.

5) Please provide the manufacturer details (model, city, or country) for all characterization instruments. In addition, it is not necessary to describe the instruments and the information derived from them.

6) It is suggested that Tables 1-3 would be combined and the Figures 2 and 3 could be removed.

7) For Figures 4 and 5, error bars are needed and the number of replicates needs to be stated. Standard deviation bars should also be considered in the discussions.

8) The manuscript needs to be thoroughly revised because it contains a few typos and errors.

Author Response

Reviewer 2 - Answers

Publication title :

“Comparison of physical and thermal properties of mulching  films made of different polymeric materials”

System number : materials-1982982

Dear Reviewer

Dear Reviewer

Thank you for preparing an in-depth review and provided comments in order to improve the quality of the publication. I agree with all the comments and suggestions posted. The answers have been included in a separate file which I am attaching .

Round 2

Reviewer 1 Report

Agree on the current form after modifcations 

Reviewer 2 Report

Dear Authors,

I have recommended the publication of your article as is.